# Medicinal Herbs and Their Derived Ingredients Protect against Cognitive Decline in In Vivo Models of Alzheimer’s Disease

**DOI:** 10.3390/ijms231911311

**Published:** 2022-09-25

**Authors:** Yueh-Ting Tsai, Shung-Te Kao, Chin-Yi Cheng

**Affiliations:** 1School of Post-Baccalaureate Chinese Medicine, College of Chinese Medicine, China Medical University, Taichung 40402, Taiwan; 2Department of Traditional Chinese Medicine, Kuang Tien General Hospital, Taichung 43303, Taiwan; 3School of Chinese Medicine, College of Chinese Medicine, China Medical University, Taichung 40402, Taiwan; 4Department of Chinese Medicine, Hui-Sheng Hospital, Taichung 42056, Taiwan; 5Department of Chinese Medicine, China Medical University Hospital, Taichung 42056, Taiwan

**Keywords:** Alzheimer’s disease, Aβ plague, medicinal herb, oxidative stress, inflammation, neuronal apoptosis

## Abstract

Alzheimer’s disease (AD) has pathological hallmarks including amyloid beta (Aβ) plaque formation. Currently approved single-target drugs cannot effectively ameliorate AD. Medicinal herbs and their derived ingredients (MHDIs) have multitarget and multichannel properties, engendering exceptional AD treatment outcomes. This review delineates how in in vivo models MHDIs suppress Aβ deposition by downregulating β- and γ-secretase activities; inhibit oxidative stress by enhancing the antioxidant activities and reducing lipid peroxidation; prevent tau hyperphosphorylation by upregulating protein phosphatase 2A expression and downregulating glycogen synthase kinase-3β expression; reduce inflammatory mediators partly by upregulating brain-derived neurotrophic factor/extracellular signal-regulated protein kinase 1/2-mediated signaling and downregulating p38 mitogen-activated protein kinase (p38 MAPK)/c-Jun N-terminal kinase (JNK)-mediated signaling; attenuate synaptic dysfunction by increasing presynaptic protein, postsynaptic protein, and acetylcholine levels and preventing acetylcholinesterase activity; and protect against neuronal apoptosis mainly by upregulating Akt/cyclic AMP response element-binding protein/B-cell lymphoma 2 (Bcl-2)-mediated anti-apoptotic signaling and downregulating p38 MAPK/JNK/Bcl-2-associated x protein (Bax)/caspase-3-, Bax/apoptosis-inducing factor-, C/EBP homologous protein/glucose-regulated protein 78-, and autophagy-mediated apoptotic signaling. Therefore, MHDIs listed in this review protect against Aβ-induced cognitive decline by inhibiting Aβ accumulation, oxidative stress, tau hyperphosphorylation, inflammation, synaptic damage, and neuronal apoptosis in the cortex and hippocampus during the early and late AD phases.

## 1. Introduction

Alzheimer’s disease (AD), a progressive age-related neurodegenerative disorder, is the most common type of dementia, accounting for 60–80% of dementia cases [1,2]. Clinical research has demonstrated that metabolic diseases such as obesity, insulin resistance, and type 2 diabetes mellitus (T2DM) and unhealthy lifestyle habits such as smoking, stress, and sleep disorders are closely associated with sporadic AD development [3,4,5]. Over 45 million people worldwide have AD, and people aged >65 years demonstrate a high prevalence of AD (7–10%), leading to death within 3–9 years after confirmed diagnosis [5,6,7,8]. AD is currently ranked as fifth leading cause of death among the elderly population in the United States [9]. Annually, the number of people with AD is increasing at a steady rate of approximately 1.8 million people worldwide, particularly in low- and middle-income countries. Therefore, medical and care costs related to AD are increasing gradually, leading to a heavy financial burden on the affected individuals’ families and the society [10,11,12]. As such, AD is becoming a global healthcare issue [13]. The behavioral characteristics of AD include a gradual decline in the quality of daily living associated with memory, learning, judgment, movement, speech, and reasoning disabilities [14,15,16]. The pathological hallmarks of AD include the extracellular accumulation of amyloid β (Aβ) plaques; intracellular aggregation of hyperphosphorylated tau (p-tau), which subsequently forms neurofibrillary tangles (NFTs); and loss of cholinergic transmission in the layer II entorhinal cortex, hippocampus, and basal forebrain—resulting in cognitive dysfunction [14,17,18,19]. The hippocampus, critical for memory and spatial learning, is vulnerable to Aβ deposition in the early stages of AD [20,21]. Aβ peptides contain 36–43 amino acids generated by the transmembrane glycoproteins expressed on cell surfaces, and Aβ_1–40_ and Aβ_1–42_ (with 40 and 42 amino acids, respectively) are the major toxic substances involved in AD pathogenesis [22,23]. In addition, Aβ_25–35_ can be generated through the enzymatic cleavage of Aβ_1–40_ [24]. Among the Aβ fragments, Aβ_25–35_ is the shortest, but has the neurotoxic properties of a full-length Aβ peptide and presents a high level of aggregation during AD pathogenesis [25]. In general, Aβ oligomer deposition, an early event in AD pathogenesis, elicits neurotoxicity, oxidative stress, synaptic dysfunction, calcium overload, inflammatory cascades, and aberrant tau hyperphosphorylation in lesions [6,26,27]. Mitogen-activated protein kinases (MAPKs) belong to a group of serine–threonine kinases; they are mainly divided into three subfamilies: extracellular regulated kinases 1/2 (ERK1/2), p38 MAPK, and c-Jun N-terminal protein kinase (JNK). These kinases are key in AD development [21]. Pharmacological enhancement of brain Aβ clearance is a potential strategy for AD prevention and treatment [28].

Donepezil, an acetylcholinesterase (AChE) inhibitor, and memantine, an N-methyl-D-aspartate receptor (NMDAR; NR) antagonist, have been approved for the clinical treatment of AD. However, these single-target drugs are not only associated with various adverse effects and toxicity but also less effective in reversing the pathological progression of AD [29,30]. By contrast, because of their multitarget and multichannel properties, medicinal herbs and their derived ingredients (MHDIs) demonstrate considerably exceptional AD treatment outcomes compared with the aforementioned drugs [11,31]. Therefore, developing potential MHDI-based AD treatment strategies is highly warranted. Here, we review the critical aspects of current studies on MHDIs for AD treatment in in vivo AD animal models and explore the potential clinical application of MHDIs in AD treatment.

## 2. MHDI-Mediated Suppression of Aβ Accumulation

Under physiological conditions, the transmembrane glycoprotein amyloid precursor protein (APP) plays a major role in central nervous system maturation and cell contact and adhesion. However, APP overexpression can cause the production of neurotoxic derivatives, closely related to AD development [32,33]. APP can be cleaved by α-secretase to initiate the nonamyloidogenic cascade preventing Aβ accumulation [34,35]. Aβ peptides are produced from APP cleavage through the activation of β- and γ-secretases in the brain regions, particularly in the temporal and frontal lobes during the early AD stages [36,37].

### 2.1. Effects of MHDIs on Aβ Accumulation through α-, β-, and γ-Secretase Activity Regulation

Notoginseng saponin Rg1, derived from *Panax notoginseng*, ameliorates cognitive deficits partly by downregulating β- and γ-secretase expression in the hippocampus at 28 days after Aβ_1–42_-induced AD [35]. In 2021, Guo et al. also reported that ginsenoside Rg1 ameliorates Aβ accumulation partly by inhibiting β-secretase in the hippocampus after 6 treatment weeks in Aβ_25–35_-induced AD [38]. Furthermore, γ-secretase is a transmembrane protein complex containing four subunits (e.g., presenilin 1), and the activity of β-site APP-cleaving enzyme-1 (BACE-1), a β-secretase, is the rate-limiting factor for Aβ accumulation, which causes hippocampal neuronal loss and cognitive dysfunction [17,35,39]. Therefore, BACE-1 may be a biomarker and therapeutic target for AD [38]. Previous studies have demonstrated that increased BACE-1 might accelerate AD pathogenesis, and pharmacological inhibition of BACE-1 reduces Aβ deposition in the brain during AD treatment [40,41]. Isorphynchophylline, extracted from *Uncaria tomentosa*, reduces Aβ generation and deposition partly through a decrease in BACE-1 expression in the brain at 129 days in TgCRND8 transgenic mice [42]. However, strong inhibition of BACE-1 causes serious adverse effects including sensorimotor gating deficits and schizophrenia, indicating that the balance of BACE-1-mediated signaling appears to be important in AD [40]. Now, researchers consider the possibility that a moderate decrease in BACE-1 activity would provide benefits and avoid adverse effects for AD prevention and treatment [40,43].

### 2.2. Summary

MHDIs mentioned in this section inhibit Aβ accumulation by upregulating α-secretase activities and downregulating β- and γ-secretase activities in the hippocampus in the late phase of AD in animal models (Table 1 and Figure 1).

## 3. MHDI-Mediated Inhibition of Aβ-Induced Oxidative Stress

Reactive oxygen species (ROS)-induced oxidative stress elicited in the early stages of AD is closely associated with Aβ generation, which leads to synaptic dysfunction and cognitive impairment [22,44]. Superoxide anions, hydroxyl radicals, and hydrogen peroxide are crucial ROS types, which attack intracellular DNA, proteins, and lipids. In addition, mitochondria are considered the main cellular source for the production of free radicals (e.g., superoxide anions) [45].

### 3.1. Involvement of Decreased Antioxidant Status and Increased Lipid Peroxidation in Aβ-Induced Oxidative Stress

Oxidative stress is caused by an imbalance between increased free radicals and decreased antioxidant enzymes such as superoxide dismutase (SOD), catalase (CAT), and glutathione peroxidase (GSH-Px). SOD catalyzes the conversion of superoxide anions to hydrogen peroxide [18]. CAT attenuates oxidative stress by converting cellular hydrogen peroxide into water and oxygen and CAT deficiency is closely related to AD pathogenesis [46]. GSH-Px can cause hydrogen peroxide clearance and diminish hydroxyl radical generation [18,47]. After Aβ accumulation, excessive ROS attack cellular organelles under impaired antioxidant defense, causing a considerable decrease in SOD, CAT, and GSH-Px levels and the exacerbation of AD progression [18,48]. ROS attack the neuronal cell membrane and then lead to neuronal cell damage through lipid peroxidation, which causes the formation of reactive aldehydes, such as 4-hydroxynonenal (4-HNE) and malondialdehyde (MDA), resulting in increased membrane permeability and decreased membrane activity [18,39]. Moreover, MDA, a toxic lipid peroxidation byproduct, disrupts protein synthesis, eventually leading to cognitive impairment [49].

### 3.2. Effects of MHDIs on Aβ-Induced Oxidative Stress through Antioxidant Activity and Lipid Oxidation Regulation

Ginsennoside Rd, derived from *Panax ginseng*, ameliorates memory and learning deficits partly by downregulating 4-HNE expression in the hippocampus at 5 days after Aβ_1–40_-induced AD [50]. Nuclear-related factor-2 (Nrf2), a pivotal transcription factor, translocates to the nucleus, binds to the antioxidant response element, produces antioxidant factors, and regulates the defense system to protect against oxidative stress [51]. Kynurenic acid can activate Nrf2-mediated signaling to reduce oxidative stress-induced neuronal damage [52]. Lignans, isolated from *Schisandra chinensis* Baill, protect against Aβ-induced oxidative stress by promoting kynurenic acid–induced Nrf2-mediated signaling in the brain at 28 days after Aβ_25–35_-induced AD [53]. GSH-Px, a major endogenous antioxidant, is key to ROS detoxification and cellular redox homeostasis maintenance. Thus, disruption of GSH-Px homeostasis in the brain is closely related to AD development [31]. However, elevated expression of GSH-Px has been noted in the brains of patients with mild cognitive deficits [54]. In 2014, Chen et al. found that bajijiasu, isolated from *Morinda officinalis*, alleviates Aβ-induced oxidative stress mainly through SOD, CAT, and GSH-Px activity upregulation and MDA activity downregulation in the hippocampus at 25 days after Aβ_25–35_-induced AD [23]. Safflower yellow, isolated from *Carthamus tinctorius*, reduces Aβ-induced oxidative stress partly by upregulating SOD and GSH-Px activities and downregulating MDA activity in the hippocampus at 28 days after Aβ_1–42_-induced AD [55]. In 2018, Zang et al. demonstrated that GJ-4, extracted from *Gardenia jasminoides* J. Ellis, alleviates memory deficit partly via increased SOD and decreased MDA levels in the cortex and hippocampus at 10 days after Aβ_25–35_-induced AD [56]. Tenuigenin, derived from *Polygala tenuifolia* Willd., effectively ameliorates memory deficit and oxidative stress mainly through increased SOD and GSH-Px activities and decreased MDA and 4-HNE activities in the hippocampus at 28 days after streptozotocin (STZ)-induced AD [47]. In 2019, Zhang et al. reported that ginsenoside Rg3, isolated from *P. ginseng* C. A. Meyer, prevents cognitive dysfunction partly by enhancing SOD, CAT, and GSH-Px expression and reducing MDA expression in the hippocampus at 60 days after D-galactose-induced AD [57]. In 2020, Yin et al. reported that neferine, isolated from *Nelumbo nucifera*, protects against cognitive deficits partly by restoring SOD, CAT, and GSH-Px activities in the hippocampus at 4 days after aluminum chloride (AlCl_3_)-induced AD [58]. However, *Rhodiola crenulata* extract was noted to alleviate oxidative stress by downregulating GSH-Px expression in the hippocampus at 28 days after Aβ_1–42_-induced AD [31]. In 2021, Shunan et al. demonstrated that betalin, from *Beta vulgaris* L., significantly attenuates cognitive deficits partly by upregulating SOD, CAT, and GSH-Px expression and downregulating MDA expression in the hippocampus at 28 days after AlCl_3_-induced AD [48].

### 3.3. Summary

Taken together, the MHDIs mentioned in this section inhibit Aβ-induced oxidative stress mainly by enhancing the activities of antioxidant enzymes such as SOD, CAT, and GSH-Px and reducing the levels of lipid peroxidation products such as 4-HNE and MDA in the cortex and hippocampus in the early and late phases of AD in animal models (Table 2 and Figure 2).

## 4. MHDI-Mediated Downregulation of Tau Hyperphosphorylation

Under physiological conditions, tau, a microtubule-associated protein, promotes the stability of axonal microtubules and regulation of axonal transport. After phosphorylation, tau detaches from microtubules and then elicits axonal transport dysfunction and synaptic toxicity [17,59]. In AD pathogenesis, tau is hyperphosphorylated and it forms paired helical filaments, which are the main constituents of NFTs [60]. Aβ-induced tau hyperphosphorylation is processed through the activation of glycogen synthase kinase-3 beta (GSK-3β), MAPKs, hyperhomocysteinemia (HHcy), and cyclin-dependent kinase 5 (CDK5); the balance between GSK-3β and protein phosphatase 2A (PP2A) activities determines the phosphorylation status of tau in an AD brain [13,17,61]. Moreover, CDK5 plays a role in the early phase of p-tau formation [55].

### 4.1. Effects of MHDIs on Aβ-Induced Tau Hyperphosphorylation through PP2A, CDK5, and GSK-3β Expression Regulation

In 2014, Yang et al. reported that *Dendrobium nobile* Lindl. attenuates p-tau by downregulating GSK-3β expression in the hippocampus at 7 days after lipopolysaccharide (LPS)-induced AD [62]. Safflower yellow attenuates tau hyperphosphorylation partly by enhancing PP2A expression and reducing CDK5 and GSK-3 expression in the hippocampus at 28 days after Aβ_1–42_-induced AD [55]. Emodin, extracted from *Rheum officinale*, significantly ameliorates HHcy-mediated Aβ-induced tau hyperphosphorylation partly by upregulating PP2A expression and downregulating BACE-1 expression in the hippocampus at 14 days after homocysteine (Hcy)-induced AD [61]. In 2019, Zhang et al. demonstrated that *R. crenulata* extract effectively ameliorates p-tau expression partly by increasing p-GSK-3β (Ser9)/GSK-3β ratio in the hippocampus at 28 days after Aβ_1–42_-induced AD [63]. In addition, *Centella asiatica* prevents tau hyperphosphorylation by downregulating GSK-3β expression and upregulating PP2A activity in the hippocampus at 10 weeks after d-galactose/AlCl_3_-induced AD [13]. According to clinical case reports, the brains of patients with AD demonstrate a considerable increase in the levels of tau phosphorylated at Thr205, Ser396, and Ser404, as well as increased GSK-3β expression but decreased PP2A expression [13,64]. In AD pathogenesis, p-tau accumulation in the hippocampal cornu ammonis 1 (CA1) occurs earlier than in other brain regions. Moreover, the extent of p-tau aggregation in the hippocampus reveals a close relationship of p-tau with cognitive function [65]. ERK1/2 and phosphoinositide 3 kinase (PI3K)/protein kinase B (Akt), which are upstream factors of GSK-3β, inhibit GSK-3β activity through the phosphorylation of GSK-3α at Ser21 and GSK-3β at Ser9, resulting in the suppression of p-tau-induced neuronal injury [17,44,66,67]. Sulforaphene, from *Raphani semen*, inhibits p-tau accumulation partly by upregulating Akt (Ser473)/GSK-3β (Ser9)–mediated signaling in the hippocampus at 6 weeks after STZ-induced AD [64]. The seeds of *Litchi chinensis* fraction ameliorate tau-induced cognitive impairment by upregulating Akt expression and downregulating GSK-3β expression in the hippocampal CA1 region at 28 days after Aβ_25–35_-induced AD [67].

### 4.2. Summary

MHDIs mentioned in this section, therefore, provide beneficial effects against tau hyperphosphorylation through the upregulation of p-Akt and PP2A expression and the downregulation of GSK-3β and CDK5 expression in the hippocampus in the early and late phases of AD in animal models (Table 3 and Figure 3).

## 5. MHDI-Mediated Reduction of Aβ-Induced Inflammation

In the early stages of AD pathogenesis, Aβ deposition–induced inflammatory responses activate microglia and astrocytes, which secrete pro-inflammatory cytokines, such as tumor necrosis factor-α (TNF-α), interleukin (IL)-1β, IL-6, inducible nitric oxide synthase (iNOS), cyclooxygenase-2 (COX-2), transforming growth factor-α, and chemokines and 5-lipoxygenase (5-LO), and thus disrupting the blood–brain barrier and exacerbating neuronal damage in the hippocampus, eventually worsening the presentation of early-stage AD [18,65,68,69,70]. Large amounts of inflammatory cytokines can activate BACE-1 and γ-secretase to suppress Aβ clearance, thereby increasing Aβ accumulation [71]. TNF-α plays a pivotal role in activation of the subsequent cytokines (IL-1β and IL-6) through various signaling pathways and then leads to the activation of nuclear factor-κB (NF-κB) [72]. 5-LO is a crucial enzyme in the formation of pro-inflammatory leukotrienes. Pharmacological inhibition of 5-LO alleviates memory deficits, synaptic dysfunction, and p-tau accumulation in a mouse model of AD [73].

### 5.1. Effects of MHDIs on Aβ-Induced Inflammation through Inflammatory Mediator Regulation

Emodin inhibits Aβ-induced inflammation by reducing TNF-α, IL-6, 5-LO, and NF-κB expression in the hippocampus at 14 days after Hcy-induced AD [61]. In 2019, Guo et al. demonstrated that ethyl acetate, extracted from *Picrasma quassioides* Benn., suppresses neuroinflammation and reduces Aβ accumulation by downregulating TNF-α, IL-1β, and IL-6 expression in the hippocampus at 23 days after Aβ_25–35_-induced AD [74]. Neferine and betalin inhibit Aβ-induced inflammation by reducing TNF-α, IL-1β, IL-6, iNOS, COX-2, and NF-κB expression in the hippocampus at 4 and 28 days after AlCl_3_-induced AD, respectively [48,58]. In 2020, Chen et al. reported that timosaponin BII, isolated from *Anemarrhena asphodeloides* Bunge, protects against cognitive impairment partly through the downregulation of TNF-α, IL-1β, and iNOS expression in the hippocampus at 38 days after LPS-induced inflammation and AD [75]. Moreover, in 2020, Song et al. found that schisandrin, derived from *S. chinensis* Baill., effectively reduces inflammatory response by downregulating TNF-α, IL-1β, IL-6, and NF-κB expression in the hippocampus at 14 days after STZ-induced AD [72]. Furthermore, cuban policosanol, purified from *Saccharum officinarum*, was noted to ameliorate amyloid plaque deposition mainly through the downregulation of TNF-α, IL-1β, and IL-6 expression in the cortex and hippocampus after 4 months in 5xFAD mice [18]. Activated microglia can be divided into two phenotypes: M1 (classical type) and M2 (alternative type). M1 microglia can secrete excessive amounts of pro-inflammatory factors and then exacerbate brain injury in AD. By contrast, M2 microglia can secrete anti-inflammatory cytokines [IL-10, IL-13, and Arginase 1 (Arg1)] and neurotrophic factors to promote repair of damaged neurons. Arg1, highly expressed in M2 microglia, can compete with iNOS for the common substrate L-arginine, resulting in reduced nitric oxide production and inflammatory damage [76,77]. Furthermore, M1 microglia triggers pro-inflammatory astrocyte activity, whereas M2 microglia promotes anti-inflammatory astrocyte activity [76]. Thus, M1-to-M2 microglia conversion contributes to synapse protection in AD hippocampus [78,79]. Ginsennoside Rd attenuates cognitive decline by downregulating ionized calcium-binding adapter molecule 1 (Iba1; a marker of microglia), glial fibrillary acidic protein (GFAP; a marker of reactive astrocytes), TNF-α, IL-1β, IL-6, and caspase-3 expression and upregulating IL-10 expression in the hippocampus at 5 days after Aβ_1–40_-induced AD [50]. Brain-derived neurotrophic factor (BDNF) is a key regulator in the synaptic plasticity contributing to the development of cognitive function, whereas in AD pathogenesis, the reduction of BDNF and insulin-like growth factor 1 (IGF-1) levels in the brain is tightly associated with Aβ accumulation and cognitive impairment [80,81]. Ginsenoside Rg5, derived from *P. ginseng*, attenuates Aβ-induced inflammation and Aβ deposition mainly by upregulating BDNF and IGF-1 expression and downregulating TNF-α, IL-1β, iNOS, and COX-2 expression in the cortex and hippocampus at 28 days after STZ-induced AD [81]. In 2019, Zhang et al. demonstrated that safflower yellow ameliorates Aβ-induced inflammation and cognitive decline mainly through the downregulation of TNF-α, IL-1β, and IL-6 expression and upregulation of Arg1 expression in the cortex and hippocampus at 20 days after Aβ_1–42_-induced AD [77].

### 5.2. Effects of MHDIs on Aβ-Induced Inflammation through Receptor for Advanced Glycation End Product- and MAPK-Mediated Signaling Regulation

The receptor for advanced glycation end product (RAGE), a pattern recognition receptor, is abundantly expressed in neurons, microglia, and astrocytes. In Aβ-induced inflammation, increased RAGE expression in neurons and glia cells leads to excessive ROS generation, which results in oxidative stress. In addition, binding of Aβ to RAGE in microglia (M1) enhances the production of pro-inflammatory cytokines, which induce NF-κB activation by modulating inflammatory signaling pathways (such as the MAPK signaling pathway). NF-κB activation, in turn, produces large amounts of pro-inflammatory cytokines, ROS, and RAGE; this results in the formation of a vicious cycle between RAGE and NF-κB, exacerbating Aβ accumulation in the cortex and hippocampus [68]. Thus, pharmacological downregulation of the activity of the inflammatory signaling pathways effectively attenuate the severity of cognitive deficits [22]. Moreover, cytokines can upregulate β-secretase activity and then augment Aβ formation [75]. Tanshinone IIA, from *Salvia miltiorrhiza* Bunge, attenuates Aβ accumulation partly by downregulating RAGE/NF-κB-mediated inflammatory signaling in the hippocampus at 30 days in APP/PS1 transgenic mice [68]. However, reactive astrocytes also play positive roles in AD in glial scar formation, limiting the extent of Aβ-induced damage, as well as in BDNF upregulation through tropomyosin receptor kinase B (TrkB) receptor expression [27]. In addition, astrocytes are important in clearing Aβ deposits [50]. MAPKs are also critical regulators of pro-inflammatory signaling response [51]. Aβ-induced inflammatory response can be elicited through the activation of JNK- and p38 MAPK-mediated signaling [82,83,84]. Cytokine release, in turn, activates p38 MAPK-mediated signaling and subsequently causes tau hyperphosphorylation [62]. In 2016, Wang et al. reported that caffeic acid, extracted from *Ocimum gratissimum*, ameliorates Aβ-induced inflammation mainly by downregulating p38 MAPK/NF-κB-mediated signaling in the hippocampus at 2 weeks after Aβ_1–40_-induced AD [85]. A steroid-enriched fraction of *Achyranthes bidentata* protects against Aβ-induced inflammation partly through the downregulation of p38 MAPK/JNK/NF-κB-mediated signaling in the cortex and hippocampus at 16 days after Aβ_1–40_-induced AD [84]. In 2021, Yamamoto et al. found that rosmarinic acid suppresses Aβ-induced inflammation by downregulating JNK-mediated signaling in the hippocampus after 8 treatment months in a triple-transgenic mouse model of AD [65]. The effects of M1-to-M2 transformation on the formation of synaptic plasticity are attributable to the activation of the BDNF/TrkB/ERK1/2-mediated signaling pathway [78]. Safflower yellow, extracted from *C. tinctorius* L., effectively enhances synaptic plasticity by activating the BDNF/TrkB/ERK1/2-mediated signaling pathway in the cortex and hippocampus at 3 months in APP/PS1 mice [78]. However, in 2019, Wang et al. reported that Aβ_1–42_ effectively enhances the ERK1/2 signaling cascade, representing a close connection between Aβ and ERK1/2-mediated signaling [86]. In addition, in 2013, Ashabi et al. indicated that the inhibition of ERK1/2 activation, p38 MAPK activation, or both attenuates neuroinflammation in an Aβ-induced AD model [51].

### 5.3. Summary

MHDIs mentioned in this section thus reduce Aβ-induced inflammation mainly by inhibiting the expression of inflammatory factors, such as TNF-α, IL-1β, IL-6, iNOS, COX-2, NF-κB, and 5-LO and promoting the expression of anti-inflammatory factors, such as IL-10 and Arg1. Furthermore, the anti-inflammatory effects of MHDIs against Aβ-induced neuronal damage are partly attributable to the upregulation of BDNF/TrkB/ERK1/2-mediated signaling and downregulation of p38 MAPK/JNK-mediated signaling in the cortex and hippocampus in the early and late phases of AD in animal models (Table 4 and Figure 4).

## 6. MHDI-Mediated Amelioration of Aβ-Induced Synaptic Dysfunction

In the brain, the mammalian target of rapamycin plays a major role in dendritic growth and synaptic plasticity development [3]. Synaptic activity is essential in synaptic plasticity and memory formation, and maintenance of synaptic activity effectively protects against AD pathogenesis [87]. The maintenance of normal synaptic plasticity requires particular proteins, including immediate early genes (IEG) and activity-regulated cytoskeleton-associated protein (Arc), which are crucial for long-term memory formation and consolidation [88]. Synaptic plasticity disruption followed by synapse loss caused by Aβ oligomers in the hippocampal CA1 subregion occurs in the early stages of AD, and the hippocampal CA1 subregion is more vulnerable to AD-related neuronal damage than are the other subregions. In addition, synapse loss and dendritic spine abnormalities are closely associated with cognitive decline [36,59,87,89].

### 6.1. Involvement of Synaptic Protein Expression in Aβ-Induced Synaptic Dysfunction

The synapse-associated proteins, involving presynaptic dynamin 1, synapsin-1 (SYN-1), synaptophysin (SYP), postsynaptic density protein (PSD)-95, and neural cell adhesion molecule, play a crucial role in synaptic plasticity and memory formation [68,90]. SYN-1, a presynaptic marker, is significantly expressed in synaptic vesicles, and it plays a crucial role in the modulation of neurotransmitter release [78]. SYN-1 expression can reflect synapse density [91]. Moreover, SYP, a calcium-binding protein, is a presynaptic vesicle protein, with a role in synaptic formation and vesicular endocytosis [78,92]. PSD-95, a critical scaffolding component of postsynaptic terminals, is vital for synaptic transmission and synaptic stabilization during long-term potentiation (LTP) [27,93]. Dendritic spine density is also crucial for synaptic function and cognitive behavior [36]. Microtubule-associated protein 2 (MAP-2), a dendritic marker, is a pivotal factor for dendritic spine development and dendritic elongation. Thus, upregulated MAP-2 expression exerts beneficial effects against synaptic dysfunction through dendritic morphology maintenance in Aβ-damaged neurons [90]. By contrast, activation of RhoA, a member of the Rho–GTPase family, and its downstream target ROCK reduces dendritic spine density and length during AD pathogenesis [94]. Moreover, the accumulation of p-tau in the hippocampus reduces MAP-2 expression, leading to cognitive dysfunction [95]. APP also has critical physiological roles in dendritic spine density and synaptic plasticity [96]. Protein kinase c (PKC)/BDNF-mediated signaling plays a key role in synaptogenesis, synapse development, synaptic transmission, and synaptic plasticity in the hippocampus and the related cortical regions in AD animal models [27,90]. PKC plays an essential role in the modulation of the survival and apoptotic pathways. Moreover, BDNF is essential for cognitive function through the regulation of axonal sprouting and synaptic plasticity [97].

### 6.2. Effects of MHDIs on Aβ-Induced Synaptic Dysfunction through Synaptic Protein Expression Regulation

In 2014, Zhan et al. reported that berberine rescues synaptic/memory deficits by upregulating IEG and Arc mRNA and protein levels in the hippocampus at 7 weeks after D-galactose-induced AD [88]. *Xanthoceras sorbifolia* extract increases dendritic spine density probably through the activation of BDNF/TrkB/PSD-95-mediated signaling and inhibition of RhoA/ROCK-mediated signaling in the hippocampus at 18 days after Aβ_25–35_-induced AD [94]. In 2017, Ji et al. reported that daucosterol palmitate, extracted from *Alpinia oxyphylla* Miq., ameliorates Aβ-induced cognitive impairment partly due to the enhancement of SYP expression in the hippocampus at 14 days after Aβ_1–42_-induced AD [92]. Catalpol, extracted from *Rehmanniae Radix*, effectively promotes the expression of synaptic proteins including dynamin 1, SYP, PSD-95, and MAP-2 by activating PKC/BDNF-mediated signaling in the hippocampus at 2 months in aged rats [90]. BDNF combines with its receptor TrkB to activate Akt/cyclic AMP response element-binding protein (CREB)-mediated signaling. Akt is the upstream regulator of CREB, which plays a key role in the maintenance of synaptic plasticity during the pathogenesis of AD [98]. However, Aβ accumulation can suppress the proteolytic cleavage of pro-BDNF, which reduces the BDNF levels [99]. Icariin, isolated from *Epimedium brevicornum* Maxim, attenuates Aβ-induced synaptic dysfunction through the activation of BDNF/TrkB/Akt/CREB-mediated signaling in the hippocampus at 28 days after Aβ_1–42_-induced AD [98]. In addition, molecular chaperones exhibit diverse functions such as protein folding and Aβ disaggregation. Thus, chaperone proteins protect against Aβ-induced synaptic injury in the hippocampal and cortical neurons by preventing Aβ oligomers binding to the dendrites [93].

### 6.3. Involvement of Acetylcholine Release in Aβ-Induced Synaptic Dysfunction

Cholinergic neurons that release acetylcholine (ACh) from axon terminals are most closely associated with cognitive function; therefore, loss of cholinergic neurons causes memory and learning deficits [11,26]. ACh synthesis and degradation require choline acetyltransferase (ChAT) and AChE, respectively. Thus, brain ACh levels can be increased by promoting ChAT function or reduced by upregulating AChE activity [100]. Aβ accumulation alters neurotransmitter-related enzyme expression and thus increases AChE activity but reduces ChAT activity, resulting in reduced synaptic transmission and plasticity [6,101]. Increased AChE levels, in turn, trigger Aβ aggregation, leading to exacerbation of Aβ accumulation [101]. In the early stages of AD, ACh neuromediator synthesis is reduced [102].

### 6.4. Effects of MHDIs on Aβ-Induced Synaptic Dysfunction through ChAT, ACh, and AChE Level Regulation

Galantamine, a phenanthrene alkaloid isolated for the first time from *Galanthus woronowii* [103], is the first nutraceutical to be approved by the United States Food and Drug Administration as a reversible AChE inhibitor [104]. Moreover, galantamine can block ACh degradation in the synaptic cleft, resulting in constant ACh stimulation of cholinergic receptors [104,105]. In 2006, Meunier et al. demonstrated that galantamine protects against Aβ-induced memory deficits partly by inhibiting AChE activity in the hippocampus at 7 days after Aβ_25–35_-induced AD [106]. Galantamine also acts as an allosteric modulator of nicotinic ACh receptors (nAChRs) [107,108]. Galantamine enhances microglial Aβ clearance partly by upregulating microglial α7 nAChR expression in the hippocampus at 2 weeks after Aβ_42_-induced AD [107]. It has been suggested that AChE plays a key role in Aβ accumulation in the early stages of senile plaque formation [109]. In 2022, Siddique et al. reported that galantamine effectively inhibits Aβ_42_ aggregation mainly by reducing AChE activity and promoting GSH-Px levels in the brain at 57 days in the transgenic *Drosophila* model of AD [109]. Currently, galantamine provides beneficial effects on mild to moderate AD by downregulating AChE activity and upregulating ACh release in the brain [110]. However, galantamine can cause some adverse effects, such as hepatotoxicity and gastrointestinal disorders, and cannot reduce the rate of decline of cognitive capacities in the later stages of AD [105,111]. *Gastrodia elata* Blume treatment significantly improves spatial memory mainly by upregulating ChAT expression and downregulating AChE expression in the prefrontal cortex and hippocampus at 52 days after Aβ_25–35_-induced AD [101]. In 2014, Huang et al. reported that bajijiasu ameliorates Aβ-induced cognitive dysfunction partly through increased ACh levels and decreased AChE levels in the hippocampus at 25 days after Aβ_25–35_-induced AD [23]. Lychee seed extract improves cognitive dysfunction probably by inhibiting Aβ, tau, and AChE formation in the hippocampus at 8 weeks in a rat model of T2DM and AD [112]. In 2018, Zang et al. observed that GJ-4 improves cognitive ability partly by downregulating AChE levels and upregulating ACh levels in the cortex and hippocampus at 10 days after Aβ_25–35_-induced AD [56]. Lignans, isolated from *S. chinensis* Baill, ameliorate cognitive decline partly through the upregulation of ACh levels in the brain at 1 week in AD rats [11].

### 6.5. Involvement of Postsynaptic Receptor and Protein Expression in Aβ-Induced Synaptic Dysfunction

NMDARs (NRs) and α-amino-3-hydroxy-5-methyl-4-isoxazole-propionicaci (AMPA) receptors (AMPARs), both belonging to ionotropic glutamate receptors, play multiple roles in synaptic plasticity and excitotoxicity [113]. NMDAR and AMPAR [including glutamate A1 (GluA1) and GluA2 subunits] are the major components of PSD, and these receptors can regulate excitatory synaptic connections and maintenance process of LTP [91]. NMDARs are ligand-gated ion channels and their subtypes, such as NR1/NR2A (NMDAR2A) and NR1/NR2B (NMDAR2B), are regulated in the synaptic transmission process [113,114]. Ca^2+^/calmodulin (CaM)-dependent kinase II (CaMKII), a multifunctional serine/threonine protein kinase, is a pivotal enzyme in Ca^2+^/CaM-mediated signaling. CaMKII isoforms are derived from four genes (α, β, γ, and δ), and CaMKIIα is important for learning and memory [115]. Under physiological conditions, NMDAR, CaMKII, and PKC in postsynaptic density are important in synaptic plasticity [32,116]. Intracellular calcium ions phosphorylate CaMKII, which subsequently activates downstream ERK/CREB-mediated signaling for the induction of LTP in the hippocampus [115]. By contrast, in AD pathogenesis, Aβ deposition triggers extracellular Ca^2+^ flow into the cytoplasm, ultimately leading to calcium overload. This calcium overload subsequently causes neurotoxicity, reducing the expression of AMPAR 1 (GluA1), CaMKII, PKC, and NR2B contained in NMDARs [89,91,116]. Thus, Aβ accumulation disturbs NMDAR-associated LTP induction by affecting NR2A/NR2B ratio in the hippocampal CA1 and dentate gyrus [113], whereas synaptic NMDAR activation causes neuroprotective effects on Aβ intraneuronal accumulation through the enhancement of synaptic activity and plasticity [59].

### 6.6. Effects of MHDIs on Aβ-Induced Synaptic Dysfunction through Postsynaptic Receptor and Protein Expression Regulation

In 2013, Wei et al. reported that β-asarone, isolated from *Acori graminei* Rhizoma, effectively alleviates cognitive decline by activating CaMKIIα/CREB-mediated signaling in the frontal cortex at 4 months in APP/PS1 mice [117]. Oleanolic acid, from *Ligustrum lucidum*, ameliorates Aβ-induced memory deficit partly by upregulating NMDAR2B, CaMKII, and PKC expression in the hippocampus at 28 days after Aβ_25–35_-induced AD [116]. However, in 2012, Liu et al. reported that pathological cytoplasmic calcium overload occurs through the activation of NR1 subunits of NMDARs. Overloaded Ca^2+^ combines with CaM to subsequently elicit increased CaMKII phosphorylation, and this in turn promotes NR1 expression; this creates a vicious cycle between the NR1 and CaMKII expression, causing neuronal cell death in the hippocampal CA1 region [113].

### 6.7. Summary

MHDIs mentioned in this section reduce Aβ-induced synapse loss and promote synaptic proteins including dynamin 1, SYP, PSD-95, and MAP-2 by activating BDNF/Akt/CREB-mediated signaling in the hippocampus. Moreover, they ameliorate synaptic transmission deficits mainly through the upregulation of ACh PKC, NR2B, and CaMKII expression and downregulation of AChE expression in the hippocampus in the early and late phases of AD in animal models (Table 5 and Figure 5).

## 7. MHDI-Mediated Attenuation of Aβ-Induced Apoptosis

In the later stages of AD, hippocampal neuronal apoptosis plays a major role in AD pathogenesis [13]. The mammalian system exhibits two major apoptotic pathways: (1) an extrinsic pathway, induced by death receptors located on the cell membrane, and (2) an intrinsic pathway, elicited via a mitochondria-related route [118].

### 7.1. Involvement of MAPK and PI3K/Akt Signaling in Aβ-Induced Apoptosis

MAPKs play different roles in signal transduction in response to various stimuli. Moreover, MAPKs are key modulators of cell growth, differentiation, development, cell survival, and apoptosis [119,120]. In general, ERK1/2-mediated signaling is involved in cell survival, proliferation, and development, whereas p38 MAPK- and JNK-mediated signaling triggers mitochondria-related neuronal apoptosis in response to various types of stress stimuli [21,25,121]. However, ERK1/2-mediated signaling also triggers Aβ-induced apoptosis in a rat model of ibotenic acid (IBO)-induced AD [122]. Aβ accumulation leads to the phosphorylation of p38 MAPK, which subsequently triggers tau hyperphosphorylation, disrupts synaptic plasticity, and eventually, elicits hippocampal neuronal apoptosis [25]. JNK is activated by the upstream factors including apoptosis signal-regulating kinase 1 (ASK1), MAPK kinase (MKK) 4 and MKK7. Activated JNK phosphorylates nuclear factors such as c-Jun and activating transcription factor 2, as well as the cytosolic substrate APP. Thus, Aβ-induced neuronal apoptosis is closely associated with JNK-mediated signaling [123]. ERK1/2 and p38 MAPK-mediated signaling activation elicits neuronal apoptosis during Aβ accumulation in the hippocampus [21,124] and JNK activation can exert anti-apoptotic effects by downregulating mitochondria-dependent caspase-3 activity in an in vitro AD model [120]. The PI3K/Akt/GSK-3β signaling pathway plays a pivotal role in cell proliferation and differentiation, neural network maintenance, as well as neuronal growth, survival and apoptosis [125]. In AD pathogenesis, Aβ deposition inhibits PI3K/Akt activation and then triggers the expression of pro-apoptotic factors such as GSK-3β and NF-κB, resulting in neuronal apoptosis [67].

### 7.2. Effects of MHDIs on Aβ-Induced Apoptosis through MAPK-, PI3K/Akt-, and BDNF/CREB-Mediated Signaling Regulation

In 2016, Zong et al. reported that icariin, the main component from *E. brevicornum* Maxim, attenuates Aβ-induced caspase-3-apoptoic cascade and improves spatial learning mainly through the downregulation of NF-κB-, ERK1/2-, p38 MAPK-, and JNK-mediated signaling in the hippocampus at 20 days after IBO-induced AD [122]. Butylphthalide exerts beneficial effects against apoptotic neuronal death probably through the downregulation of p38 MAPK-mediated signaling in the hippocampus at 30 days after Aβ_1–42_-induced AD [10]. In 2020, Zhou et al. reported that *Tinospora sinensis* protects against AD-induced neuronal damage partly by upregulating PI3K/Akt-mediated anti-apoptotic signaling in the hippocampus at 21 days after Aβ_1–40_-induced AD [125]. Aβ-induced neuronal apoptosis can be suppressed through the activation of BDNF/TrkB-mediated signaling, which subsequently leads to CREB phosphorylation, enabling memory preservation [29]. The translocation of phosphorylated CREB to the nucleus induces the transcription of anti-apoptotic factors including B-cell lymphoma 2 (Bcl-2) and B-cell lymphoma-extra large (Bcl-xL) [126]. Icariside II, extracted from *E. brevicornum*, mitigates Aβ-induced apoptotic neuronal death by activating BDNF/TrkB/CREB-mediated signaling in the hippocampus at 5 days after Aβ_25–35_-induced AD [29].

### 7.3. Involvement of Mitochondria-Mediated Apoptotic Cascades in Aβ-Induced Apoptosis

MAPKs regulate apoptotic signaling through the modulation of Bcl-2 family members in AD animal models [44]. Bcl-2 family proteins, including pro-apoptotic proteins [i.e., Bcl-2-associated x protein (Bax), Bcl-2 antagonist killer 1 (Bak), and Bcl-2-associated death promoter (Bad)] and anti-apoptotic proteins (i.e., Bcl-2, Bcl-xL, and myeloid cell leukemia 1), rigorously regulate mitochondrial outer membrane (MOM) integrity and permeability [13]. Bcl-2 could bind to Bax (Bak) and then prevent Bax (Bak) translocation to the MOM during the apoptotic process [127]. Moreover, Bcl-2 and Bax (Bak) play a pivotal role in the regulation of the mitochondrial permeability transition pore [57]. Thus, the balance between pro-apoptotic and anti-apoptotic Bcl-2 determines whether cells survive or undergo apoptosis [128,129]. JNK-mediated signaling induces Bax translocation from the cytosol to MOM, leading to the disruption of MOM integrity and the induction of mitochondria-related apoptotic protein release into the cytosol [130,131,132]. Mitochondria-related apoptosis can occur via caspase-dependent or -independent pathways. In mitochondria-related apoptotic signaling, MOM permeabilization causes the release of cytochrome c (cyt c) into the cytosol, where it binds to apoptosis protease-activating factor 1 in the presence of dATP and then forms the apoptosome, leading to the activation of caspase-9/caspase-3 (final apoptosis executor)-mediated apoptosis. In addition, MOM integrity disruption causes mitochondrial apoptosis-inducing factor (AIF) release into the cytosol and then translocation to the nucleus, resulting in caspase-independent apoptosis [15]. Bcl-2 family proteins are pivotal regulators of mitochondria-related apoptotic cascades [133].

### 7.4. Effects of MHDIs on Aβ-Induced Apoptosis through Bax-, Cullin 4B-, and β-Catenin-Mediated Signaling Regulation

β-Asarone, the major ingredient of *Acorus tatarinowii* Schott, protects against apoptotic neuronal death partly by downregulating JNK-mediated Bax/caspase-9 signaling in the hippocampus at 28 days after Aβ_1–42_-induced AD [119]. In 2016, Wang et al. demonstrated that genistein exerts beneficial effects against cognitive deficits probably by downregulating Bax/cyt c/caspase-3-mediated apoptotic signaling in the hippocampus at 49 days after Aβ_25–35_-induced AD [15]. In 2018, Wei et al. found that 2-dodecyl-6-methoxycyclohexa-2, 5-diene-1, 4-dione (DMDD), from *Averrhoa carambola* L., protects against Aβ-induced apoptosis mainly by increasing Bcl-2/Bax expression and suppressing cleaved caspase-9 and caspase-3 expression in the hippocampus at 21 days in APP/PS1 transgenic AD mice [133]. Scutellarein, derived from *Scutellaria baicalensis*, effectively reduces Aβ-induced apoptosis mainly through the upregulation of Bcl-2 expression and downregulation of Bax and cleaved caspase-3 expression in the hippocampus at 28 days after Aβ-induced AD [134]. Ginsenoside Rg3 attenuates Aβ-induced mitochondria-related apoptosis through the downregulation of Bax/caspase-9/caspase-3- and Bax/AIF-mediated apoptotic signaling in the hippocampus at 60 days after D-galactose-induced AD [57]. Aβ and p-tau accumulation have a close connection with mitochondria-mediated cyt c/caspase-3 apoptosis [120,135]. Somatostatin receptor 4 (SSTR4), mainly distributed in the cortex and hippocampus, is crucial in learning and memory. SSTR4 can upregulate Aβ-degrading enzyme expression in the hippocampus, whereas ubiquitin-mediated degradation of SSTR4 initiates neuronal apoptosis in AD pathogenesis [8]. Cullin 4B (CUL4B) overexpression promotes SSTR4 ubiquitination, resulting in AD exacerbation [8,136]. In 2020, Weng et al. demonstrated that tetramethylpyrazine, extracted from *Ligusticum wallichii*, attenuates cognitive impairment by downregulating CUL4B/SSTR4-mediated apoptotic signaling in the hippocampus at 30 days in APP/PS1 mice [8]. The Wnt/β-catenin signaling pathway plays crucial roles in regulation of cell proliferation, migration, and differentiation. Activation of Wnt/β-catenin-mediated signaling reduces Aβ-induced caspase-3-mediated apoptosis in the hippocampus in AD pathogenesis [137]. In 2018, Xie et al. indicated that SOX8, a high mobility group-box transcription factor, could activate the Wnt/β-catenin signaling pathway in an in vitro cell culture model [138]. Notoginsenoside R2, extracted from *P. notoginseng*, alleviates Aβ-induced caspase-3-related apoptosis by activating SOX8/β-catenin-mediated signaling in the hippocampus at 20 weeks after Aβ_25–35_-induced AD [139].

### 7.5. Effects of MHDIs on Aβ-Induced Apoptosis through Endoplasmic Reticulum Stress and Autophagy Signaling Regulation

The endoplasmic reticulum is a cell organelle for protein synthesis and translation. The deposition of misfolded proteins in the endoplasmic reticulum causes endoplasmic reticulum stress (ERS), which promotes apoptotic cell death and is closely associated with AD occurrence. ERS also leads to apoptosis mainly through C/EBP homologous protein (CHOP) and glucose-regulated protein 78 (GRP78) activation and caspase-12-mediated signaling in the cortex and hippocampus [140]. Crocin, extracted from *Crocus sativus* L., protects against Bax/caspase-3-mediated apoptosis probably by downregulating GRP78 (an ERS marker) and CHOP expression in the prefrontal cortical neurons and also the hippocampal CA1 region at 14 days after Aβ_25–35_-induced AD [140]. In 2020, Song et al. reported that schisandrin effectively ameliorates ERS mainly through the downregulation of CHOP, GRP78, and cleaved caspase-12 expression in the hippocampus at 14 days after STZ-induced AD [72]. ERS can induce autophagy via different signaling pathways [141]. Autophagy, an essential process for scavenging damaged cells and misfolded proteins, plays a crucial role in reducing Aβ deposition [135]. In the autophagic process, the microtubule-associated protein 1 light chain 3 (LC3), including cytosolic type I (LC3-I) and membrane bound type II (LC3-II), is the crucial component of the autophagosomal membrane; and autophagic activity is positively associated with the LC3-II/LC3-I ratio [142]. In addition, Beclin-1 regulates an early step in autophagosomal membrane formation [143]. In the early stages of AD, increased autophagy leads to reduced Aβ accumulation, whereas autophagy inhibition exacerbates it. Aβ secretion and deposition are regulated through autophagy [135]. However, in the later stages of AD, overexpression of autophagy proteins triggers toxic oligomeric Aβ release. Thus, maintaining autophagic flux homeostasis is a potential strategy for AD treatment [144]. Euxanthone, extracted from *Polygala caudate*, attenuates Aβ-induced apoptotic neuronal death by upregulating Bcl-2/Bax and LC3B-II expression in the hippocampus at 16 days after Aβ_1–42_-induced AD [145]. However, in 2019, Jiang et al. demonstrated that icariin protects from Aβ-induced apoptosis partly by downregulating Beclin-1, LC3-II/LC3-I, and cleaved caspase-3 expression in the hippocampus at 5 days after Aβ_1–42_-induced AD [144].

### 7.6. Summary

MHDIs mentioned in this section exert neuroprotective effects against Aβ-induced apoptosis mainly by upregulating PI3K/Akt/CREB/Bcl-2-mediated anti-apoptotic signaling and downregulating p38 MAPK/JNK/Bax/caspase-3- and Bax/AIF-mediated apoptotic signaling in the hippocampus. Moreover, the anti-apoptotic effects of MHDIs are partly due to the modulation of CUL4B/SSTR4-, SOX8/β-catenin-, CHOP/GRP78-, and LC3-II/LC3-I/Bectin-1-mediated signaling in the cortex and hippocampus in the early and late phases of AD in animal models (Table 6 and Figure 6).

## 8. Conclusions

In AD pathogenesis, Aβ oligomer deposition elicits oxidative stress, tau hyperphosphorylation, inflammatory cascades, synapse loss, and neuronal apoptosis. MHDIs listed in this review can suppress Aβ accumulation mainly through β- and γ-secretase activity downregulation. The antioxidative stress effects of MHDIs are mainly due to the enhancement of antioxidant activities such as SOD, CAT, and GSH-Px and reduction in lipid peroxidation. Moreover, MHDIs effectively prevent tau hyperphosphorylation by upregulating PP2A expression and downregulating GSK-3β expression. MHDIs reduce inflammatory mediators such as TNF-α, IL-1β, IL-6, iNOS, COX-2, NF-κB, and 5-LO partly through the upregulation of BDNF/ERK1/2-mediated signaling and downregulation of p38 MAPK/JNK-mediated signaling. In addition, MHDIs attenuate synapse loss and synaptic transmission deficits mainly by increasing dynamin 1, SYP, PSD-95, MAP-2, and ACh levels but decreasing AChE levels. Furthermore, MHDIs protect against neuronal apoptosis mainly through upregulation of Akt/CREB/Bcl-2- and SOX8/β-catenin-mediated anti-apoptotic signaling and downregulation of p38 MAPK/JNK/Bax/caspase-3-, Bax/AIF-, CUL4B-, CHOP/GRP78-, and autophagy-mediated apoptotic signaling. In summary, MHDIs listed in this review exert neuroprotective effects against Aβ-induced cognitive decline by downregulating Aβ accumulation, oxidative stress, tau hyperphosphorylation, inflammation, synaptic damage, and neuronal apoptosis in the cortex and hippocampus in the early and late phases in in vivo models of AD. Therefore, MHDIs listed in this review probably exhibit multitarget and multichannel properties in AD treatment.

## 9. Future Directions

In AD pathogenesis, Aβ-induced synaptic dysfunction, inflammation, and apoptosis in the cortex and hippocampus are main pathological responses in worsening AD. Therefore, further research for the development of potential restorative MHDI-based clinical therapeutic strategies for AD is warranted.

## Figures and Tables

**Figure 1 ijms-23-11311-f001:**
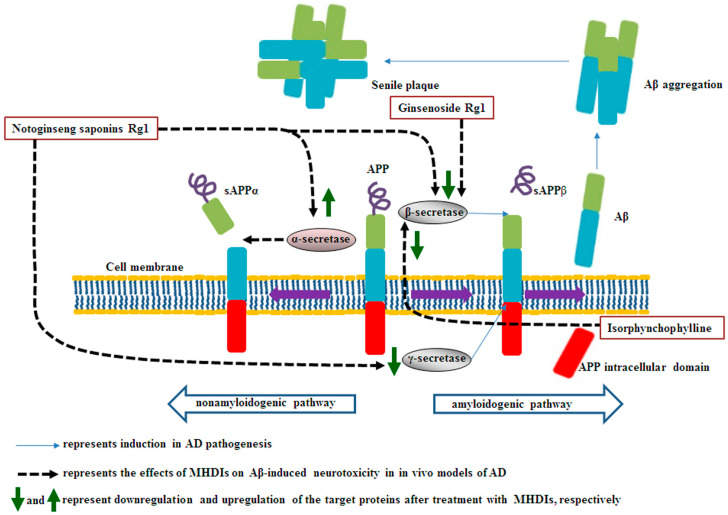
Schematic representation of the effects of MHDIs on Aβ accumulation in the hippocampus in the late phase of AD in in vivo models. sAPP, soluble amyloid precursor protein.

**Figure 2 ijms-23-11311-f002:**
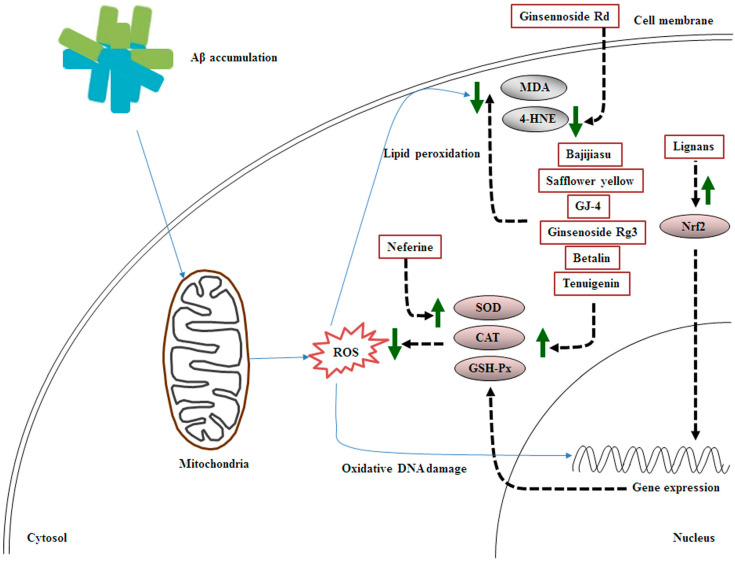
Schematic representation of the effects of MHDIs on Aβ-induced oxidative stress in the cortex and hippocampus in the early and late phases of AD in in vivo models.

**Figure 3 ijms-23-11311-f003:**
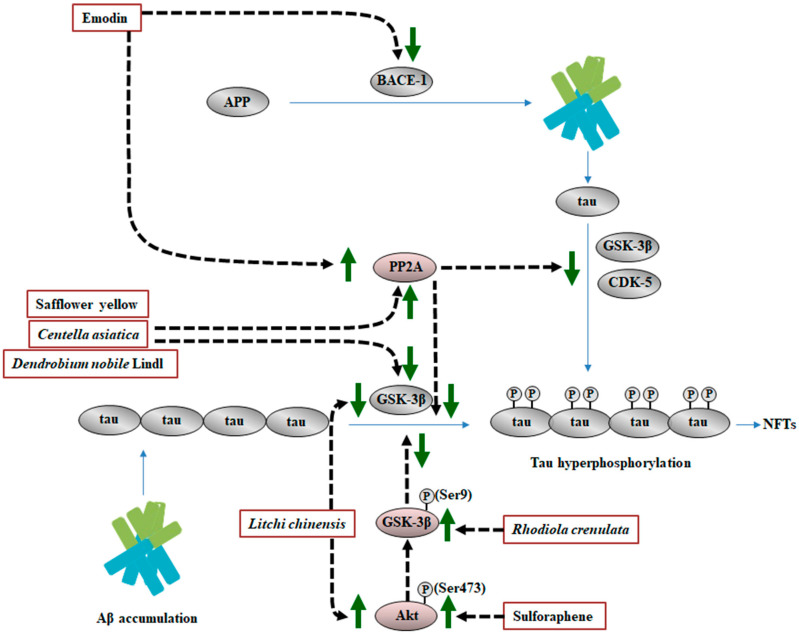
Schematic representation of the effects of MHDIs on Aβ-induced tau hyperphosphorylation in the hippocampus in the early and late phases of AD in in vivo models. P, phosphorylated.

**Figure 4 ijms-23-11311-f004:**
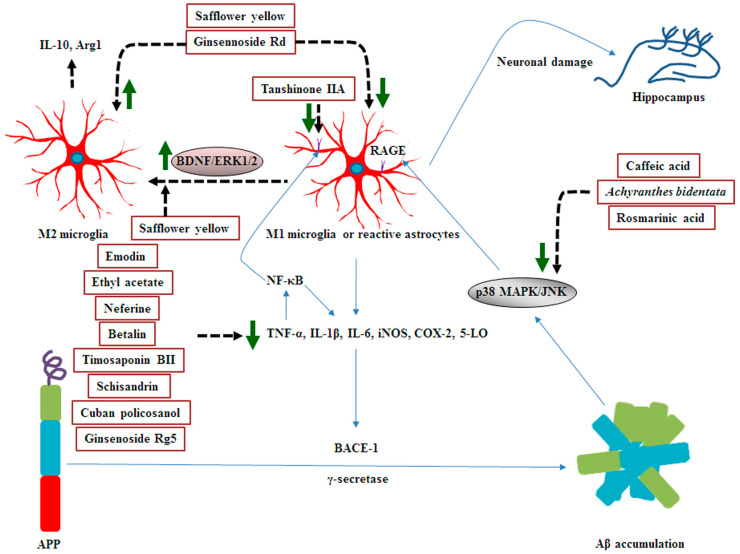
Schematic representation of the effects of MHDIs on Aβ-induced inflammation in the cortex and hippocampus in the early and late phases of AD in in vivo models.

**Figure 5 ijms-23-11311-f005:**
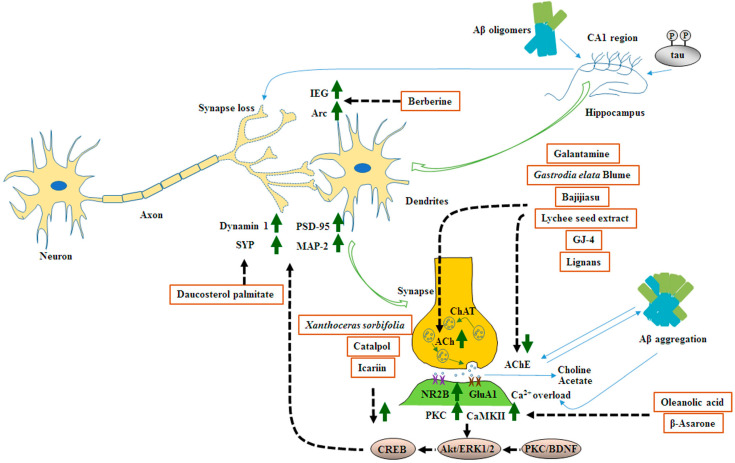
Schematic representation of the effects of MHDIs on Aβ-induced synaptic dysfunction in the hippocampus in the early and late phases of AD in in vivo models.

**Figure 6 ijms-23-11311-f006:**
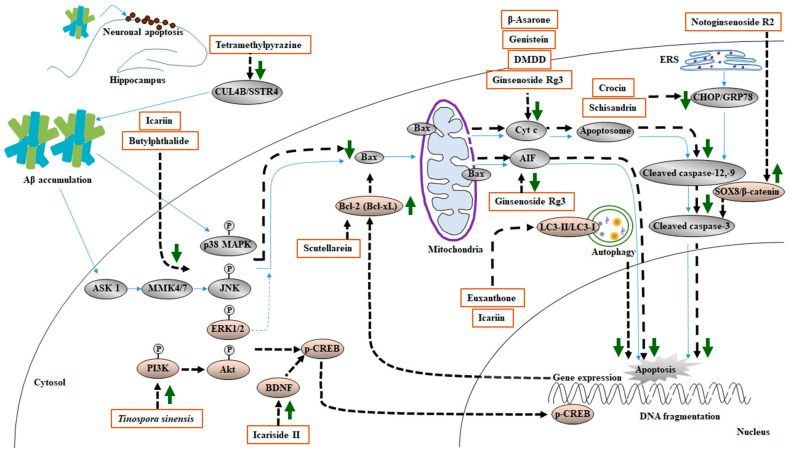
Schematic representation of the effects of MHDIs on Aβ-induced apoptosis in the cortex and hippocampus in the early and late phases of AD in in vivo models.

**Table 1 ijms-23-11311-t001:** MHDIs that suppress Aβ accumulation in AD animal models.

Major Ingredients	Isolated from Medicinal Herbs	Anti-Aβ Accumulation Activities	Models	Reference
Notoginseng saponin Rg1	*Panax notoginseng*	α-secretase↑, β- secretase↓, γ-secretase↓	28 days after Aβ_1–42_-induced AD	[35]
Ginsenoside Rg1		Bcl-2↑, MAP-2↑, NeuN↑, Bax↓, β-secretase↓	6 weeks after Aβ_25–35_-induced AD	[38]
Isorphynchophylline	*Uncaria tomentosa*	BACE-1↓, presenilin 1↓, p-APP (Thr668) ↓	129 days in TgCRND8 transgenic mice	[42]

Bcl-2, B-cell lymphoma 2; MAP-2, microtubule-associated protein 2; NeuN, neuronal nuclei, Bax, Bcl-2-associated x protein.

**Table 2 ijms-23-11311-t002:** MHDIs that inhibit Aβ-induced oxidative stress in AD animal models.

Major Ingredients	Isolated from Medicinal Herbs	Antioxidative Stress Activities	Models	References
Ginsennoside Rd	*Panax ginseng*	4-HNE↓	5 days after Aβ_1–40_-induced AD	[50]
Lignans	*Schisandra chinensis* Baill	kynurenic acid↑, Nrf2↑	28 days after Aβ_25–35_-induced AD	[53]
Bajijiasu	*Morinda officinalis*	SOD↑, CAT↑, GSH-Px↑, MDA↓	25 days after Aβ_25–35_-induced AD	[23]
Safflower yellow	*Carthamus tinctorius*	SOD↑, GSH-Px↑, MDA↓	28 days after Aβ_1–42_-induced AD	[55]
GJ-4	*Gardenia jasminoides* J. Ellis	SOD↑, MDA↓, iNOS↓, COX-2↓, PGE2↓, TNF-α↓	10 days after Aβ_25–35_-induced AD	[56]
Tenuigenin	*Polygala tenuifolia* Willd	SOD↑, GSH-Px↑, MDA↓, 4-HNE↓ p-tau (Ser396) ↓, p-tau (Thr181) ↓	28 days after STZ-induced AD	[47]
Ginsenoside Rg3	*P. ginseng* C. A. Meyer	SOD↑, CAT↑, GSH-Px↑, MDA↓,	60 days after D-galactose-induced AD	[57]
Neferine	*Nelumbo nucifera*	SOD↑, CAT↑, GSH-Px↑	4 days after AlCl_3_-induced AD	[58]
	*Rhodiola crenulata*	GSH-Px↓, arachidonic acid↓	28 days after Aβ_1–42_-induced AD	[31]
Betalin	*Beta vulgaris* L.	SOD↑, CAT↑, GSH-Px↑, MDA↓,	28 days after AlCl_3_-induced AD	[48]

iNOS, inducible nitric oxide synthase; COX-2, cyclooxygenase-2; PGE2, prostaglandin E2; TNF-α, tumor necrosis factor-α.

**Table 3 ijms-23-11311-t003:** MHDIs that downregulate tau hyperphosphorylation in AD animal models.

Major Ingredients	Isolated from Medicinal Herbs	Anti-p-Tau Activities	Models	References
	*Dendrobium nobile* Lindl.	GSK-3β↓ p-tau (Ser199-202) ↓, p-tau (Ser396) ↓, p-tau (Ser404) ↓, p-tau (Thr231), p-tau (Thr205)	7 days after LPS-induced AD	[62]
Safflower yellow	*C. tinctorius*	PP2A↑, CDK5↓, GSK-3↓	28 days after Aβ_1–42_-induced AD	[55]
Emodin	*Rheum officinale*	PP2A↑, p-CREB↑, SYP↑, SYN-1↑, BACE-1↓,	14 days after Hcy-induced AD	[61]
	*Centella asiatica*	PP2A↑, Bcl-2 mRNA↑, GSK-3β↓	10 weeks after d-galactose/AlCl_3_-induced AD	[13]
	*R. crenulata*	GSK-3β (Ser9)/GSK-3β↑	28 days after Aβ_1–42_-induced AD	[63]
Sulforaphene	*Raphani semen*	p-Akt (Ser473) ↑, p-GSK-3β (Ser9) ↑, IL-10↑, TNF-α↓, IL-6↓	6 weeks after s STZ-induced AD	[64]
	Seed of Litchi chinensis	Akt↑, GSK-3β↓	28 days after Aβ_25–35_-induced AD	[67]

CREB, cyclic AMP response element-binding protein; SYP, synaptophysin; SYN-1, synapsin-1; STZ, streptozotocin.

**Table 4 ijms-23-11311-t004:** MHDIs that reduce Aβ-induced inflammation in AD animal models.

Major Ingredients	Isolated from Medicinal Herbs	Anti-Inflammation Activities	Models	References
Emodin	*R. officinale*	microglia activation↓, TNF-α↓, IL-6↓, 5-LO↓, NF-κB↓	14 days after Hcy-induced AD	[61]
Ethyl acetate	*Picrasma quassioides* Benn	TNF-α↓, IL-1β↓, IL-6↓	23 days after Aβ_25–35_-induced AD	[74]
Betalin	*B. vulgaris* L.	TNF-α mRNA↓, IL-1β mRNA↓, IL-6 mRNA↓, iNOS mRNA↓, COX-2 mRNA↓, NF-κB↓	28 days after AlCl_3_-induced AD	[48]
Neferine	*N. nucifera*	TNF-α↓, IL-1β↓, IL-6↓, iNOS↓, COX-2↓, NF-κB↓	4 days after AlCl_3_-induced AD	[58]
Timosaponin BII	*Anemarrhena asphodeloides* Bunge	TNF-α↓, IL-1β↓, iNOS↓	38 days after LPS-induced inflammation and AD	[75]
Schisandrin	*S. chinensis* Baill	Sirtuin 1↑, TNF-α↓, IL-1β↓, IL-6↓, NF-κB↓	14 days after STZ-induced AD	[72]
Cuban policosanol	*Saccharum officinarum*	4-HNE↓, TNF-α↓, IL-1β↓, IL-6↓	4 months in 5xFAD transgenic mice	[18]
Ginsennoside Rd	*Panax ginseng*	IL-10↑, HSP70↑, Iba1↓, GFAP↓, TNF-α↓, IL-1β↓, IL-6↓, caspase-3↓	5 days after Aβ_1–40_-induced AD	[50]
Ginsenoside Rg5	*P. ginseng*	BDNF↑, IGF↑, ChAT↑, TNF-α↓, IL-1β↓, iNOS↓, COX-2↓, AChE↓	28 days after STZ-induced AD	[81]
Safflower yellow	*C. tinctorius* L.	TNF-α↓, IL-1β↓, IL-6↓, iNOS mRNA↓, Arg1↑(marker of M2 microglia), YM-1 mRNA↑ (M2-related cytokine), CD206 mRNA↑ (M2-related cytokine)	28 days after Aβ_1–42_-induced AD	[77]
Tanshinone IIA	*salvia miltiorrhiza* Bunge	TNF-α↓, IL-1β↓, IL-6↓, RAGE↓, NF-κB↓	30 days in APP/PS1 transgenic mice	[68]
Caffeic acid	*Ocimum gratissimum*	p-p38 MAPK↓, NF-κB-p65↓, TNF-α↓, IL-6↓, p53↓, AChE↓, CAT↑, GSH-Px↑	14 days after Aβ_1–40_-induced AD	[85]
	*Achyranthes bidentata*	p-p38 MAPK↓, p-JNK↓ TNF-α↓, IL-1β↓, IL-6↓	16 days after Aβ_1–40_-induced AD	[84]
Rosmarinic acid		p-JNK↓, p-c-Jun↓	8 months in the triple-transgenic mouse model of AD	[65]
Safflower yellow	*C. tinctorius* L.	Arg1↑, BDNF ↑, TrkB ↑, p-ERK1/2↑ iNOS↓	3 months in APP/PS1 transgenic mice	[78]

ChAT, choline acetyltransferase.

**Table 5 ijms-23-11311-t005:** MHDIs that ameliorate Aβ-induced synaptic dysfunction in AD animal models.

Major Ingredients	Isolated from Medicinal Herbs	Restoring Synaptic Dysfunction Activities	Models	References
Berberine		IEG mRNA & protein↑, Arc mRNA & protein↑	7 weeks after D-galactose-induced AD	[88]
	*Xanthoceras sorbifolia*	PSD-95↑, BDNF↑, p-TrkB/TrkB↑, RhoA↓, ROCK2↓	18 days after Aβ_25–35_-induced AD	[94]
Daucosterol palmitate	*Alpinia oxyphylla* Miq.	SYP↑	14 days after Aβ_1–42_-induced AD	[92]
Catalpol	*Rehmanniae Radix*	dynamin 1↑, SYP↑, PSD-95↑, MAP-2↑	2 months in aged rats (23–24 months old)	[90]
Icariin	*Epimedium brevicornum* Maxim	PSD-95↑, BDNF↑, TrkB↑, Akt↑, CREB↑	28 days after Aβ_1–42_-induced AD	[98]
Galantamine	*Galanthus woronowii*	AChE↓	7 days after Aβ_25–35_-induced AD	[106]
Galantamine		microglial α7 nAChR↑	2 weeks after Aβ_42_-induced AD	[107]
Galantamine		AChE↓, GSH-Px↑, caspase-9 activity↓, caspase-3 activity↓	57 days in the transgenic *Drosophila* model of AD	[109]
	*Gastrodia elata* Blume	ChAT↑, AChE↓	52 days after Aβ_25–35_-induced AD	[101]
Bajijiasu	*Morinda officinalis*	ACh↑, AChE↓	25 days after Aβ_25–35_-induced AD	[23]
Lychee seed extract	*Litchi chinensis*	AChE↓	8 weeks in a rat model of T2DM and AD	[112]
GJ-4	*G. jasminoides* J. Ellis	ACh↑, AChE↓	10 days after Aβ_25–35_-induced AD	[56]
Lignans	*S. chinensis* Baill	ACh↑	1 week in AD rats	[11]
β-Asarone	*Acori graminei Rhizoma*	CaMKIIα↑, p-CREB↑, Bcl-2↑	4 months in APP/PS1 mice	[117]
Oleanolic acid	*Ligustrum lucidum*	NMDAR2B↑, CaMKII↑, PKC↑, BDNF↑, TrkB↑, CREB↑	28 days after Aβ_25–35_-induced AD	[116]

**Table 6 ijms-23-11311-t006:** MHDIs that attenuate Aβ-induced apoptosis in AD animal models.

Major Ingredients	Isolated from Medicinal Herbs	Anti-Apoptotic Activities	Models	References
Icariin	*E. brevicornum* Maxim	Bcl-2/Bax↑, NF-κB↓,p-ERK1/2/ERK1/2↓, p-p38 MAPK/p38 MAPK↓, p-JNK/JNK↓	20 days after IBO-induced AD	[122]
Butylphthalide		p38 MAPK mRNA & protein↓	30 days after Aβ_1–42_-induced AD	[10]
	*Tinospora sinensis*	p-PI3K/PI3K↑, p-Akt/Akt↑	21 days after Aβ_1–40_-induced AD	[125]
Icariside II	*E. brevicornum* Maxim	BDNF↑, TrkB↑, p-CREB/CREB↑	5 days after Aβ_25–35_-induced AD	[29]
β-asarone	*Acorus tatarinowii* Schott	ASK 1↓, p-MKK7↓, p-c-Jun↓, Bad mRNA & protein↓, Bax mRNA & protein↓, cleaved caspase-9 mRNA & protein↓	28 days of Aβ_1–42_-induced AD	[119]
Genistein		Bax↓, cyt c↓, caspase-3↓	49 days after Aβ_25–35_-induced AD	[15].
DMDD	*Averrhoa carambola* L.	Bcl-2/Bax↑, cleaved caspase-9↓, cleaved caspase-3↓	21 days in APP/PS1 transgenic AD mice	[133]
Scutellarein	*Scutellaria baicalensis*	Bcl-2↑, Bax↓, caspase-3↓, nucleus NF-κB↓	28 days after Aβ-induced AD	[134]
Ginsenoside Rg3	*P. ginseng* C. A. Meyer	Bcl-2↑, Bax↓, caspase-9↓, caspase-3↓, AIF↓	60 days after D-galactose-induced AD	[57]
Tetramethylpyrazine	*Ligusticum wallichii*	SSTR4↑, CUL4B↓	30 days in APP/PS1 transgenic mice	[8]
Notoginsenoside R2	*P. notoginseng*	SOX8↑, β-catenin↑, cleaved caspase-3↓, COX-2↓	20 weeks after Aβ_25–35_-induced AD	[139]
Crocin	*Crocus sativus* L.	GRP78↓, CHOP↓, Bax↓, caspase-3↓	14 days after Aβ_25–35_-induced AD	[140]
Schisandrin	*S. chinensis* Baill	GRP78↓, CHOP↓, cleaved caspase-12↓	14 days after STZ-induced AD	[72]
Euxanthone	*Polygala caudate*	Bcl-2/Bax↑, LC3B-II↑	16 days after Aβ_1–42_-induced AD	[145]
Icariin	*E. brevicornum* Maxim	p-Akt↑, LC3-II/LC3-I↓, Beclin-1↓, Cathepsin D (neurofibrillary degeneration marker) ↓	5 days after Aβ_1–42_-induced AD	[144]

## Data Availability

Not applicable.

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
