# Peer review of "Medicinal Herbs and Their Derived Ingredients Protect against Cognitive Decline in In Vivo Models of Alzheimer’s Disease"

_ijms, 2022, doi:10.3390/ijms231911311_

Round 1

Reviewer 1 Report

Present manuscript has numerous grammatical and syntax errors. The article needs rigorous revision for English language.

Author Response

Present manuscript has numerous grammatical and syntax errors. The article needs rigorous revision for English language.

ANS.: The original manuscript has been rigorously checked and corrected to minimize grammatical and syntax errors.

Reviewer 2 Report

In this paper the authors describe the effects of medical herbs as potential treatment of AD. The paper is well written and cover the possible mechanisms through which medical herbs may be useful in the clinical practice. I also really appreciate the tables and the figures which well summerize the main aspect of each paragraph. 

In my opinion, the paper can be accepted in the present form

Author Response

We greatly appreciate the reviewer’s comments.

Reviewer 3 Report

This review attempts to provide an overview on the effects of medicinal herbs and their derived ingredients (MHDIs) that have been tested in animal models of Alzheimer's disease. A very large number of publications have been evaluated although the list is not complete (as one example: galantamine is not mentioned). The potential sites of interference of the MHDIs are very well illustrated in the figures and grouped according to pathways related to A-beta accumulation, oxidative stress, tau, inflammation, synapse, and apoptosis.

The term "medicinal herbs and their derived ingredients (MHDIs)" is helpful as a designation of the many diverse substances, however, it is not acceptable to extrapolate the action of the single substances to the entire group (as one example last sentence of abstract: Therefore, MHDIs protect against Aβ-induced cognitive decline by inhibiting Aβ accumulation, oxidative stress, tau hyper- phosphorylation, inflammation, synaptic damage, and neuronal apoptosis in the cortex and hippo- campus during the early and late AD phases.)

Very often, the original references are not cited but are referred to by publications that eventually mention the original findings (hear-say) but are not the original work, therefore, these citations are not appropriate. To provide just one example: "AD is divided into two major forms: sporadic and familial; moreover, most cases are sporadic, whereas in only a small proportion of cases, AD is inherited as an autosomal dominant condition [1, 2]"

  1. Sadigh-Eteghad, S.; Sabermarouf, B.; Majdi, A.; Talebi, M.; Farhoudi, M.; Mahmoudi, J. Amyloid-beta: a crucial factor in Alzheimer's disease. Med. Princ. Pract. 2015, 24, 1-10. 
  2. Nobakht, M.; Hoseini, S. M.; Mortazavi, P.; Sohrabi, I.; Esmailzade, B.; Rahbar Rooshandel, N.; Omidzahir, S., Neuropathological changes in brain cortex and hippocampus in a rat model of Alzheimer's disease. Iran. Biomed. J. 2011, 15, 51-58 

These papers surely do not provide the primary evidence for the above statement.

Numerous statements are overexaggerated or overinterpreted, the effects observed are not necessarily causally connected and have not been shown to have any relation with the human disease. As only one example: "Notoginseng saponins Rg1, derived from Panax notoginseng, ameliorates spatial learning and memory deficits by upregulating α-secretase expression and downregulat- ing β- and γ-secretase expression in the hippocampus at 28 days after Aβ1-42-induced AD [34]" if it was shown that spatial learning is influenced and that alpha secretase is upregulated it is not appropriate to conclude that the upregulation of alpha secretase is causal to ameliorate spatial learning. Coincidence does not prove causality!

Furthermore, parts of the literature are ignored. The discussion of BACE (page 3) is only one example where opposing evidence is not mentioned (e.g. reviewed by McDade et al. 2021) and further work by R. Vassar, L. Mucke labs and more is not considered.

Author Response

We greatly appreciate the reviewer’s comments. The manuscript has been revised and compiled as the reviewer’s suggestions. Our point-by-point responses to the reviewer’s comments are described as follows:

  1. This review attempts to provide an overview on the effects of medicinal herbs and their derived ingredients (MHDIs) that have been tested in animal models of Alzheimer's disease. A very large number of publications have been evaluated although the list is not complete (as one example: galantamine is not mentioned). The potential sites of interference of the MHDIs are very well illustrated in the figures and grouped according to pathways related to A-beta accumulation, oxidative stress, tau, inflammation, synapse, and apoptosis.

ANS.: The anti-Alzheimer’s disease effect of galantamine has been mentioned in the MHDI-Mediated Amelioration of Aβ-Induced Synaptic Dysfunction section on p.15 lines 30-32 in the revised manuscript, and the descriptions are “Galantamine is a phenanthrane alkaloid extracted from Galanthus woronowii [103]. In 2006, Meunier et al. demonstrated that galantamine protects against Aβ-induced memory deficits partly by inhibiting AChE activity in the hippocampus at 7 days after Aβ25-35-induced AD [104]”. The effect of galantamine on Aβ-induced synaptic dysfunction has also been mentioned in Table 5 and Figure 5.

References

  1. Yuede, C. M.; Dong, H.; Csernansky, J. G. Anti-dementia drugs and hippocampal-dependent memory in rodents. Behav. Pharmacol. 2007, 18, 347-363.
  2. Meunier, J.; Ieni, J.; Maurice, T. The anti-amnesic and neuroprotective effects of donepezil against amyloid beta25-35 peptide-induced toxicity in mice involve an interaction with the sigma1 receptor. Br. J. Pharmacol. 2006, 149, 998-1012.

  1. The term "medicinal herbs and their derived ingredients (MHDIs)" is helpful as a designation of the many diverse substances, however, it is not acceptable to extrapolate the action of the single substances to the entire group (as one example last sentence of abstract: Therefore, MHDIs protect against Aβ-induced cognitive decline by inhibiting Aβ accumulation, oxidative stress, tau hyper- phosphorylation, inflammation, synaptic damage, and neuronal apoptosis in the cortex and hippo- campus during the early and late AD phases.)

ANS.: The term “MHDIs” has been corrected as “MHDIs listed (mentioned) in this review (section)” on p.1 line 17, p.3 line 25, p.6 line 5, p.8 line 24, p.12 line 4, p.17 line 1, p.21 line 29 p.23 lines 3-4, p.23 line 16, and p.23 lines 19-20 in the revised manuscript.

  1. Very often, the original references are not cited but are referred to by publications that eventually mention the original findings (hear-say) but are not the original work, therefore, these citations are not appropriate. To provide just one example: "AD is divided into two major forms: sporadic and familial; moreover, most cases are sporadic, whereas in only a small proportion of cases, AD is inherited as an autosomal dominant condition [1, 2]"
  2. Sadigh-Eteghad, S.; Sabermarouf, B.; Majdi, A.; Talebi, M.; Farhoudi, M.; Mahmoudi, J. Amyloid-beta: a crucial factor in Alzheimer's disease. Med. Princ. Pract. 2015, 24, 1-10. 
  3. Nobakht, M.; Hoseini, S. M.; Mortazavi, P.; Sohrabi, I.; Esmailzade, B.; Rahbar Rooshandel, N.; Omidzahir, S., Neuropathological changes in brain cortex and hippocampus in a rat model of Alzheimer's disease. Iran. Biomed. J. 2011, 15, 51-58 

These papers surely do not provide the primary evidence for the above statement.

ANS.: The statementAD is divided into two major forms: sporadic and familial; moreover, most cases are sporadic, whereas in only a small proportion of cases, AD is inherited as an autosomal dominant condition” has been deleted in the revised manuscript.

  1. Numerous statements are overexaggerated or overinterpreted, the effects observed are not necessarily causally connected and have not been shown to have any relation with the human disease. As only one example: "Notoginseng saponins Rg1, derived from Panax notoginseng, ameliorates spatial learning and memory deficits by upregulating α-secretase expression and downregulat- ing β- and γ-secretase expression in the hippocampus at 28 days after Aβ1-42-induced AD [34]"if it was shown that spatial learning is influenced and that alpha secretase is upregulated it is not appropriate to conclude that the upregulation of alpha secretase is causal to ameliorate spatial learning. Coincidence does not prove causality!

ANS.: In this manuscript, the literature citations have been carefully reviewed and the statements have been corrected to avoid exaggeration and over-interpretation.   

  1. Furthermore, parts of the literature are ignored. The discussion of BACE (page 3) is only one example where opposing evidence is not mentioned (e.g. reviewed by McDade et al. 2021) and further work by R. Vassar, L. Mucke labs and more is not considered.

ANS.: The discussion of BACE-1 has been mentioned on p.3 lines 14-16 and p.3 lines 18-23 in the revised manuscript, and the descriptions are “Previous studies have demonstrated that increased BACE-1 might accelerate AD pathogenesis, and pharmacological inhibition of BACE-1 reduces Aβ deposition in the brain during AD treatment [40, 41]” and “However, strong inhibition of BACE-1 causes serious adverse effects including sensorimotor gating deficits and schizophrenia, indicating that the balance of BACE-1-mediated signaling appears to be important in AD [40]. Now, researchers consider the possibility that a moderate decrease in BACE-1 activity would provide benefits and avoid adverse effects for AD prevention and treatment [40, 43]”, respectively.

References

  1. Vassar, R.; Kovacs, D. M.; Yan, R.; Wong, P. C. The beta-secretase enzyme BACE in health and Alzheimer's disease: regulation, cell biology, function, and therapeutic potential. J. Neurosci. 2009, 29, 12787-12794.
  2. Neumann, U.; Ufer, M.; Jacobson, L. H.; Rouzade-Dominguez, M. L.; Huledal, G.; Kolly, C.; Luond, R. M.; Machauer, R.; Veenstra, S. J.; Hurth, K.; et al. The BACE-1 inhibitor CNP520 for prevention trials in Alzheimer's disease. EMBO Mol. Med. 2018, 10, e9316.
  3. McDade, E.; Voytyuk, I.; Aisen, P.; Bateman, R. J.; Carrillo, M. C.; De Strooper, B.; Haass, C.; Reiman, E. M.; Sperling, R.; Tariot, P. N.; et al. The case for low-level BACE1 inhibition for the prevention of Alzheimer disease. Nat. Rev. Neurol. 2021, 17, 703-714.

Round 2

Reviewer 3 Report

The authors responded to my concerns and partially addressed the criticism. However, they missed the chance to add substantial information. As one example, galantamine is now mentioned superficially but it is not mentioned that galantamine is the major approved drug for Alzheimer's treatment and substantial studies on galantamine effects in AD models are also not cited. 

Author Response

The authors responded to my concerns and partially addressed the criticism. However, they missed the chance to add substantial information. As one example, galantamine is now mentioned superficially but it is not mentioned that galantamine is the major approved drug for Alzheimer's treatment and substantial studies on galantamine effects in AD models are also not cited. ANS.: The substantial information of galantamine and the substantial studies on galantamine effects in AD models have been added to the MHDI-Mediated Amelioration of Aβ-Induced Synaptic Dysfunction section on p.15-16 lines 30-6 in the revised manuscript, and the descriptions are “Galantamine, a phenanthrene alkaloid isolated for the first time from Galanthus woronowii [103], is the first nutraceutical to be approved by the United States Food and Drug Administration as a reversible AChE inhibitor [104]. Moreover, galantamine can block ACh degradation in the synaptic cleft, resulting in constant ACh stimulation of cholinergic receptors [104, 105]. In 2006, Meunier et al. demonstrated that galantamine protects against Aβ-induced memory deficits partly by inhibiting AChE activity in the hippocampus at 7 days after Aβ25-35-induced AD [106]. Galantamine also acts as an allosteric modulator of nicotinic ACh receptors (nAChRs) [107, 108]. Galantamine enhances microglial Aβ clearance partly by upregulating microglial α7 nAChR expression in the hippocampus at 2 weeks after Aβ42-induced AD [107]. It has been suggested that AChE plays a key role in Aβ accumulation in the early stages of senile plaque formation [109]. In 2022, Siddique et al. reported that galantamine effectively inhibits Aβ42 aggregation mainly by reducing AChE activity and promoting GSH-Px levels in the brain at 57 days in the transgenic Drosophila model of AD [109]. Currently, galantamine provides beneficial effects on mild to moderate AD by downregulating AChE activity and upregulating ACh release in the brain [110]. However, galantamine can cause some adverse effects, such as hepatotoxicity and gastrointestinal disorders, and cannot reduce the rate of decline of cognitive capacities in the later stages of AD [105, 111]”. The effects of galantamine on Aβ-induced synaptic dysfunction have also been mentioned in Table 5. References 103. Yuede, C. M.; Dong, H.; Csernansky, J. G. Anti-dementia drugs and hippocampal-dependent memory in rodents. Behav. Pharmacol. 2007, 18, 347-363. 104. Hassan, N. A.; Alshamari, A. K.; Hassan, A. A.; Elharrif, M. G.; Alhajri, A. M.; Sattam, M.; Khattab, R. R. Advances on Therapeutic Strategies for Alzheimer's Disease: From Medicinal Plant to Nanotechnology. Molecules 2022, 27, 4839. 105. Tuzimski, T.; Petruczynik, A., Determination of Anti-Alzheimer's Disease Activity of Selected Plant Ingredients. Molecules 2022, 27, 3222. 106. Meunier, J.; Ieni, J.; Maurice, T. The anti-amnesic and neuroprotective effects of donepezil against amyloid beta25-35 peptide-induced toxicity in mice involve an interaction with the sigma1 receptor. Br. J. Pharmacol. 2006, 149, 998-1012. 107. Takata, K.; Kitamura, Y.; Saeki, M.; Terada, M.; Kagitani, S.; Kitamura, R.; Fujikawa, Y.; Maelicke, A.; Tomimoto, H.; Taniguchi, T.; et al. Galantamine-induced amyloid-β clearance mediated via stimulation of microglial nicotinic acetylcholine receptors. J. Biol. Chem. 2010, 285, 40180-40191. 108. Nikiforuk, A.; Potasiewicz, A.; Kos, T.; Popik, P. The combination of memantine and galantamine improves cognition in rats: The synergistic role of the alpha7 nicotinic acetylcholine and NMDA receptors. Behav. Brain Res. 2016, 313, 214-218. 109. Siddique, Y. H.; Naz, F.; Rahul; Varshney, H. Comparative study of rivastigmine and galantamine on the transgenic Drosophila model of Alzheimer's disease. Curr. Res. Pharmacol. Drug Discov. 2022, 3, 100120. 110. Seo, E. J.; Fischer, N.; Efferth, T. Phytochemicals as inhibitors of NF-kappaB for treatment of Alzheimer's disease. Pharmacol. Res. 2018, 129, 262-273. 111. Sahoo, A. K.; Dandapat, J.; Dash, U. C.; Kanhar, S. Features and outcomes of drugs for combination therapy as multi-targets strategy to combat Alzheimer's disease. J. Ethnopharmacol. 2018, 215, 42-73.